# Genetic Analyses of Mungbean [*Vigna radiata* (L.) Wilczek] Breeding Traits for Selecting Superior Genotype(s) Using Multivariate and Multi-Traits Indexing Approaches

**DOI:** 10.3390/plants12101984

**Published:** 2023-05-15

**Authors:** Mohammad Golam Azam, Mohammad Amir Hossain, Umakanta Sarker, A. K. M. Mahabubul Alam, Ramakrishnan M. Nair, Rajib Roychowdhury, Sezai Ercisli, Kirill S. Golokhvast

**Affiliations:** 1Pulses Research Centre, Bangladesh Agricultural Research Institute, Ishurdi, Pabna 6620, Bangladesh; 2Department of Genetics and Plant Breeding, Faculty of Agriculture, Bangladesh Agricultural University, Mymensingh 2202, Bangladesh; 3Department of Genetics and Plant Breeding, Faculty of Agriculture, Bangabandhu Sheikh Mujibur Rahman Agricultural University, Gazipur 1706, Bangladesh; 4Pulses Research Sub-Station, Bangladesh Agricultural Research Institute, Gazipur 1701, Bangladesh; 5AVRDC-The World Vegetable Center, South Asia, Hyderabad 502324, India; 6Department of Biotechnology, Visva-Bharati Central University, Santiniketan 731235, India; 7Department of Horticulture, Faculty of Agriculture, Ataturk University, 25240 Erzurum, Türkiye; 8HGF Agro, Ata Teknokent, TR-25240 Erzurum, Türkiye; 9Siberian Federal Scientific Center of Agrobiotechnology RAS, 2b Centralnaya Street, Krasnoobsk 630501, Russia

**Keywords:** genetic diversity, Shannon diversity index, PCA, cluster analysis, mungbean, yield, yield-related traits

## Abstract

Mungbean [*Vigna radiata* (L.) Wilczek] is an important food, feed, and cash crop in rice-based agricultural ecosystems in Southeast Asia and other continents. It has the potential to enhance livelihoods due to its palatability, nutritional content, and digestibility. We evaluated 166 diverse mungbean genotypes in two seasons using multivariate and multi-traits index approaches to identify superior genotypes. The total Shannon diversity index (SDI) for qualitative traits ranged from moderate for terminal leaflet shape (0.592) to high for seed colour (1.279). The analysis of variances (ANOVA) indicated a highly significant difference across the genotypes for most of the studied traits. Descriptive analyses showed high diversity among genotypes for all morphological traits. Six components with eigen values larger than one contributed 76.50% of the variability in the principal component analysis (PCA). The first three PCs accounted for the maximum 29.90%, 15.70%, and 11.20% of the total variances, respectively. Yield per plant, pod weight, hundred seed weight, pod length, days to maturity, pods per plant, harvest index, biological yield per plant, and pod per cluster contributed more to PC1 and PC2 and showed a positive association and positive direct effect on seed yield. The genotypes were grouped into seven clusters with the maximum in cluster II (34) and the minimum in cluster VII (10) along with a range of intra-cluster and inter-cluster distances of 5.15 (cluster II) to 3.60 (cluster VII) and 9.53 (between clusters II and VI) to 4.88 (clusters I and VII), suggesting extreme divergence and the possibility for use in hybridization and selection. Cluster III showed the highest yield and yield-related traits. Yield per plant positively and significantly correlated with pod traits and hundred seed weight. Depending on the multi-trait stability index (MTSI), clusters I, III, and VII might be utilized as parents in the hybridization program to generate high-yielding, disease-resistant, and small-seeded mungbean. Based on all multivariate-approaches, G45, G5, G22, G55, G143, G144, G87, G138, G110, G133, and G120 may be considered as the best parents for further breeding programs.

## 1. Introduction

Mungbean [*Vigna radiata* (L.) Wilczek] is a highly nutritious, early maturing, and broad-spectrum adaptable grain legume crop majorly cultivated in South Asia, Southeast Asia, and Australia [1]. It is also highly demanded in western countries as a plant-based protein source [2]. It is readily digested, with 22–28% of seed protein, 1–1.5% fat, and 60–65% carbohydrates, as well as different vitamins, minerals, and various antioxidants [3]. Due to its nutritional value, mungbean is being exploited in many ways according to local taste demands. It is commonly eaten as “daal” soup, but may be processed to make noodles, porridge, curries, ice cream, cakes, bean paste, soups, sweets, and flour [4]. The seeds may be consumed as a dry bean, sprouting gram, or split daal in everyday meals in India and Bangladesh [5], or as vegetable bean sprouts [1]. Mungbean seeds, fodder, and haulms may also be utilized as fertilizer and animal feed [6]. Moreover, it requires minimum input of water and fertilizer and can grow in harsh environmental conditions over a broad range of temperate and tropical climates [7]. However, due to limited breeding efforts in Asia, there is a huge knowledge gap in major agronomic traits such as seed size, shape, and colour. Those are important for varietal improvement; therefore, these grain quality traits may be a vital area for research and breeding initiatives [8].

Despite its tremendous agricultural advantages and utilization, mungbean is grown on only 7 million hectares in the tropics and subtropics (8.5% of the world’s pulse area), with an annual production and productivity of 3 million tons and 721 kg per hectare, respectively [9]. Mungbean is a prominent and essential pulse crop in Bangladesh and its productivity remains low as compared to other mungbean-producing countries due to several constraints, such as low yield, poor crop management practices, variable growth habits, pod shattering, lodging, late/indeterminate maturity, vulnerability to diseases and pests, and importantly the grain quality [10]. Hence, there is an urgent need for better production of mungbean by introducing new cultivars and improving cultural techniques.

Owing to the poor genetic base of this crop, additional genetic resources must be examined to broaden the genetic diversity [8]. Many papers have been published on genetic diversity by using a limited number of genotypes in previous studies from Bangladesh [11]. The main aspect of the mungbean improvement effort is the lack of genetic variability and diversity in the primary gene pool, which may provide a wonderful chance for plant breeders to develop new and better cultivars with desirable traits [12]. Phenotypic diversity evaluation, through characterizing morphological and agronomical traits, plays a crucial role in the selection of appropriate parents for genetic improvement and future breeding programs [13,14]. In crop improvement programs, the selection of superior genotypes entirely depends on the variability of genotypes [15]. The extent of genetic erraticism [16] and the extent of heritability of desirable traits determine the success of the improvement of crop breeding [17]. Nair et al. [18] indicated that future development in mungbean breeding required urgent efforts to discern mungbean genotypes with strong agronomic traits for further improvement in research programs. Therefore, a significant level of genetic diversity is necessary to satisfy mungbean breeding goals.

As a result, extensive statistical trials, such as genetic divergence, cluster analysis, and principal component analysis (PCA), are necessary for the characterization of genetic diversity in mungbean [8]. These approaches are highly important for choosing potential genotypes for a future breeding effort. These approaches are used for grouping a large number of genotypes into homogeneous groups and determining the genetic distance across and within clusters, and principal component analysis can be used for determining the most contributing characteristics to the genetic diversity and identifying better mungbean genotypes. The variability of the qualitative and quantitative traits of crops is of great interest to researchers in developing a new variety [19,20,21,22]. However, little study has been conducted on the qualitative and quantitative traits of mungbean for its domestication worldwide [23]. Because of the complexity of legume yield-controlling mechanisms [24,25], the multi-trait stability index (MTSI) provides a unique selection process that is easily interpretable and free from weighting coefficients and multi-collinearity issues. The performance of the MGIDI index is assessed through a Monte Carlo simulation study, where the percentage of success in selecting traits with desired gains is compared with classical and modern breeding indexes under different agronomic conditions [26]. However, most of these studies applied a limited number of genotypes to fewer quantitative traits. Moreover, there is no appropriate information on the genetic diversity of mungbean genotypes through multivariate and MTSI analysis. For mungbean, a systematic, large-scale inquiry is necessary to create promising or improved cultivars. It will assist plant breeders to discover suitable genotypes as parental sources to provide a diversified population for selection and the development of improved mungbean cultivars. Therefore, the present study was performed for the trait characterization of a large number of qualitative and quantitative traits and to estimate the amount of genetic diversity, and also to select the superior genotypes, based on critical features for further mungbean improvement programs utilizing advanced multi-disciplinary breeding approaches.

## 2. Results

One hundred and sixty-six lines of mungbean germplasm were characterized for yield-contributing traits. These germplasms were collected from different agro-ecological regions of Bangladesh as well as Southeast Asia. The collection contained 39 local landraces, 39 advanced lines, and 8 modern varieties. Eighty germplasms were brought from aboard (WVC, Thailand, China, India, Australia, and Pakistan) (Appendix A).

### 2.1. Distribution Frequency and Diversity Index for Qualitative Traits

A considerable amount of natural variation was identified among the studied mungbean genotypes for the 15 analysed qualitative traits provided in Table 1 with descriptor states, phenotypic variability, and frequency %. Traits, such as seed colour, seedling vigour, leaf pubescence, stem pubescence, and seed size, among other qualitative features, were demonstrated to exhibit large diversity.

The estimation of Shannon–Weaver Diversity Indices (SDI) or H’ ranged from 0.592 (terminal leaflet shape) to 1.279 (seed colour). Intermediate phenotypic diversity (H’ = 0.50–0.75) was exhibited by hypocotyl colour, stem colour, leaf colour, leafiness, terminal leaflet shape, and calyx colour, and the rest of the traits indicated high phenotypic diversity (H’ ≥ 0.75).

### 2.2. Distribution Frequency, Pattern (Skewness and Kurtosis), and Diversity Index for Quantitative Traits

All the quantitative traits of the studied mungbean genotypes showed continuous distribution on the histogram and suggested a significant range of phenotypic variation among the genotypes (Figure 1). Descriptive statistics including mean, maximum, and minimum range, coefficient of variation (CV), data distribution pattern (skewness and kurtosis), and Shannon diversity index (SDI) for all 166 genotypes are presented in Table 2. The coefficient of variation values of quantitative traits demonstrated a high level of variation in the mungbean (Table 2). Most of the qualitative traits varied significantly in different accessions of the mungbean genotypes. A low level of variability for most of the quantitative traits was evident from a low value of CV for most of the characters studied. DFF (4.24%), DM (1.89%), PH (1.22%), CT (2.79%), SV (1.66%), PPP (3.67%), PL (1.49%), and BYPP (2.40%) showed a low level of morphological variations. However, a moderate level of morphological variation was observed for CPP (12.69%), SPP (9.15%), YPP (11.19%), and HI (10.81%). A high level of variability was observed for DF (33.76%) and ShWtPP (17.28%). All the quantitative characteristics evaluated in the germplasm demonstrated skewness values between −0.5 and 0.5, except for DFF (1.21), DF (1.29), DM (1.93), PPP (0.90), PPC (0.88), ShWtPP (0.98), YPP (1.53), and HI (1.30), which showed normal distribution in the population. Likewise, all of the quantitative traits exhibited kurtosis > 0, except for CT (−1.37), SWtPP (−0.97), PL (−0.19), HSW (−0.99), SYPP (−060), and BYPP (−0.69). The Shannon–Weaver diversity indices (SDI) for the studied 19 quantitative traits were varied (H > 0.5) for all descriptors, ranging from 4.97 for YPP to 5.11 for DF, DM, PH, and SPP. Additionally, the evenness result likewise indicated a high variation index value for all of the characteristics and ranged from 0.86 to 1.00. Similarly, the highest values of E_H_ (1.00) were recorded for DFF, DF, DM, and SPP, whereas the lowest E_H_ values (0.86) were for YPP. The average SDI for quantitative characteristics was greater than 5.07, and for the studied qualitative traits was 0.81. Considering seed size, small-sized grain was produced in 44 genotypes (26.51%), medium-sized grain in 102 genotypes (61.45%), and bold-sized grain in 20 genotypes (12.05%).

### 2.3. Analysis of Variance (ANOVA) for the Quantitative Traits

The pooled analysis of variance (ANOVA) for the two seasons’ quantitative traits demonstrated significant variation across the evaluated variables (Table 3). Significant differences (*p* ≤ 0.001, 0.01, or 0.05) were observed in genotype, years, and genotype-by-year interaction (G × Y) for all the studied variables. Only a few variables were different, for example, ShWtPP, SWtPP, and HI reported non-significant variations for G × Y. There was significant variation (*p* < 0.01) for all of the studied traits, which also revealed a possible degree of diversification among the genotypes (Appendix A). Mean performance for the different quantitative traits was presented as means and range (Appendix A). The knowledge of the degree of genetic and phenotypic variation in local landraces and the extent of association among traits is important to provide the basis for successful choice.

### 2.4. Correlation between the Quantitative Traits

Correlation analysis describes the relationship among the quantitative traits. Pearson correlation coefficients (r) for the quantitative traits are presented in Figure 2. The highest positive significant correlations were discovered between YPP and HI (r = 0.94, *p* < 0.001), followed by YPP and PPP (r = 0.80, *p* < 0.001) and PPP and HI (r = 0.79, *p* < 0.001). The most significant negative correlation was observed between DM and SPWt (r =−0.47, *p* < 0.001), followed by DM and PL (r = −0.42, *p* < 0.001) and DM and SPP (r = −0.40, *p* < 0.001). Similarly, DF revealed a high association with DM (r = 0.66, *p* < 0.001) and PH (r = 0.37, *p* < 0.001), but other variables indicated a significant negative relationship (Appendix A). The DM revealed a significant negative relationship with all the variables except for PH, although PPC and YMV had shown a non-significant effect, respectively. HSW shows a large positive relation with PWt, PL, and SPP and a strong negative correlation with DM. The YLD and HI demonstrated a significant positive correlation with PPP, HSW, SPWt, PL, and SPP, whereas other parameters showed significant negative interactions with DF and DM.

### 2.5. Principal Components Analysis (PCA) for the Association of Genotypes and Phenotypes

Results also indicated that in PC1, positive values were given for SWtPP, YPP, PWt, HSW, PL, HI, and DM (Figure 3A). SYPP, PPP, HI, YPP, BYPP, CT, and PPC were the major contributors to the observed variance in PC2 (Figure 3B). In the third principal component, DF, DFF, PH, HSW, SWtPP, and DM demonstrated higher contributions to overall morphological diversity (Figure 3C).

The score and loading plots of PCA on the major investigated characteristics of mungbean genotypes are presented in (Figure 4). All genotypes were successfully separated in all quadrants using the first two components (Figure 4A). This distribution of genotypes gives a clear indication that they represent the phenotypic diversity among the genotypes and explains how they are widely distributed along both the axes. The PCA (Figure 4B) performed on quantitative data demonstrated that the first two dimensions (PC1 and PC2) accounted for the highest variances of the overall variance. The contribution of individual compounds to sample differentiation is displayed as a correlation circle (Figure 4B) where normalized vectors graphically reflect the quantitative variables. In PC1, DFF, DF, DM, CT, and SY contributed positively, whereas the rest of the features contributed negatively (Figure 4B). In PC2, SY, HSW, YPP, PL, SPWt, and BYPP contributed to positive variance, while DM, DFF, DF, PH, CT, PPP, PPC, YPP, and HI contributed negatively (Figure 4B). YPP, HI, and PPP traits contributed adversely in both PCA dimensions. The length and direction of the vectors were substantially associated with the significance of each variable. A positive relation between compounds is larger when the angle between their directions is smaller (close to 0°), while the correlation is negative if the angle reaches 180 degrees. No linear dependence develops if the angle is appropriately fixed at 90 degrees.

Five variables showed lower magnitude with shorter vector lengths, i.e., YMV, SV, ShWtPP, PH, and CPP, whereas YPP, HI, and SY, with longer vector lengths, showed a higher magnitude (more variance) than the rest of the variables. Over the 166 verified mungbean genotypes, the traits PL, SPWt, SWtPP, and HSW were positively related to each other because they had minor or closer vector angels to each other and negatively correlated to DM, DFF, and DF, and five of these features were revealed with greater magnitude because they had the smallest vector length of all the traits, so they contributed more to the total variance (Figure 4B). YLD, HI, and PPP were considered to have strong associations, as demonstrated by the narrow angle between them in the figure. The opposite direction of the characteristics arrow suggests a negative association between them, as shown by the SY.

The Appendix A provides a comparison of eigenvalues and eigenvectors for the ten principal components among 166 mungbean germplasm, and Figure 5A,B display a scree plot created for 19 agronomic characteristics of the ten principal components. The scree plot of the PCA (Figure 5A) indicated that the first two eigenvalues correspond to the main proportion of the variance in the dataset. The graph demonstrated that the greatest variance was seen in PC1, with the highest eigenvalue of 5.7, followed by PC2 (3), PC3 (2.1), PC4 (1.5), and PC5 (1.2). The first six PCs accounted for about 76.50% of the overall variance among the genotypes with eigenvalues larger than unity for all the 19 variables examined (Appendix A and Figure 3B) (Appendix A and Figure 5B). These values were rated as met according to Kaiser’s criterion (eigenvalue > 1) [27]. Individually, the first four PCAs accounted for 64.60% of the total variance, for which PC1 showed 29.90% of the variation, while PC2, PC3, and PC4 displayed 15.70%, 11.20%, and 7.90% of the overall variation in parenthesis, respectively (Appendix A and Figure 5B).

### 2.6. PCA Biplot Analysis

In the PCA biplot, all mungbean genotypes were distributed throughout all quadrants of the PCA ellipse plot and displayed diverse clustering (Figure 6A,B); however, accessions belonging to particular varietal groups largely occupied particular quadrants based on quantitative attributes. Based on yield attributes G13, G82, G29, G128, G139, G81, G53, and G128, the genotype in cluster II (Figure 6A,B) was strongly associated with YPP, HI, and PPP, while the rest of the genotypes in cluster II were correlated with HSW, SPP, SWtPP, and SPWt, indicating that these traits might be essential for developing high-yielding mungbean production, whereas most of the genotypes in cluster I and cluster VII demonstrated a reverse relationship with yield, indicating that they were low-yielding genotypes. The genotypes in cluster V were related to DFF, DF, DM, and CT, showing that they were early or late maturing.

### 2.7. Genetic Relationship of Mungbean Genotypes through Cluster Analysis

The elbow technique was used to determine the optimal number of clusters before carrying out the cluster analysis, which indicates the variation within the groupings (within-cluster sum of a square) and shows that an optimal number of clusters is seven (Figure 7).

### 2.8. Cluster Analysis

Based on the quantitative traits, mungbean genotypes were grouped into seven distinct groups (clusters) at a 0.409 dissimilarity coefficient of variation (Appendix A and Figure 8). Clustering tree (dendrogram) indicated that similar genotypes tend to cluster together in the same group. The highest number of genotypes (#34) was grouped in cluster II, followed by cluster IV (#33) and cluster III (#28). Clusters VI, I, and V consist of 24, 20, and 17 genotypes, respectively. Cluster VII has the smallest number of genotypes (#10).

### 2.9. Intra- and Inter-Cluster Distance (D^2^)

The mean intra- and inter-cluster (D^2^) values with their corresponding intra- and inter-cluster distance are given in Table 4. The intra-cluster distance varied from 3.60 to 5.15. The highest intra-cluster (D^2^) distance was recorded for cluster II (5.15), followed by cluster IV (5.03) and cluster V (4.74). The lowest values of intra-cluster distance were reported in cluster VII (3.60), indicating the presence of fewer different genotypes grouped in this cluster. The inter-cluster distance of genotypes ranged from 4.88 to 9.53. The largest inter-cluster distance was between clusters II and VI (9.53), followed by clusters IV and VI (8.77). This indicates that crossing among these clusters generates a high and possible heterotic group. The smallest inter-cluster distance observed between I and VII (4.88) revealed genetic similarity between clusters.

### 2.10. Cluster Means Analysis

Cluster analysis combines a huge number of genotypes into small numbers of homogeneous clusters, which in turn enhances the identification of various accessions. The average performances of 19 quantitative traits in seven clusters are shown in Table 5. The highest days to flowering and days to maturity were recorded in cluster VII, followed by cluster VI and the lowest mean was seen in cluster II. Cluster VII had the highest cluster mean for plant height (77.64) and CT (27.25), and exhibited the lowest mean values for SPAD value, SPP, SPWt, YPP, and HI. Cluster I had the maximum SV, SYPP, and SPAD value, followed by cluster VI. Cluster II generated the highest cluster mean for the characteristic ShWtPP. Cluster III had the highest cluster mean value for the characteristics PPP, CPP, SPP, SPWt, SWtPP, PL, HSW, YMV, YPP, and HI used as experimental variables. Cluster IV had the lowest cluster mean values found for the characteristic CT and the highest mean value recorded in BYPP. In HSW, the lowest mean value was shown in cluster V followed by cluster VII and cluster I. More YMV-tolerant genotypes were found in clusters VI, VII, and V. Clusters III and IV comprised more disease-susceptible accessions than those of the first three clusters. In the case of DF and DM, the genotypes G3, G4, G5, G6, G12, G13, G17, G19, G24, G27, G35, G38, G46, G49, G55, G56, G59, G61, G128, and G139 were selected due to earliness (Appendix A). The genotypes G114, G115, G116, G120, G122, G123, G124, G130, G141, G149, G150, G151, and G152 of cluster V showed the lowest mean value for PH and can be selected for short stature. Cluster III contained 28 genotypes (G10, G18, G25, G26, G31, G36, G40, G51, G52, G68, G69, G72, G74, G76, G77, G79, G80, G81, G87, G93, G96, G97, G98, G99, G105, G108, G110, and G113), which showed the maximum mean value for most of the examined traits. Based on statistical analysis, the genotypes from cluster III might be considered as the best parents for PPP, CPP, PL, HSW, and YPP as high-yielding promising lines and would be used as distance parents for a hybridization program.

### 2.11. Multi-Trait Stability Index (MTSI) for Identifying Superior Mungbean Genotype(s)

Based on the data analysis, a very strong genotypic effect was observed for 19 quantitative traits presented in (Appendix A). However, the genotypes selected using the MTSI index were G12, G45, G46, G143, G139, G50, G44, G49, G4, G106, G5, G163, G103, G47, G22, G144, G107, G87, G19, G55, G3, G138, G164, G34, G110, G133, G78, G131, G79, G85, G17, G37, and G128 (Figure 9A,B). These accessions represent the superior mungbean materials in terms of high stability and overall performance within the investigated genotypes. The mean of the selected genotypes (Xs) was larger than the original mean (Xo) which contained all 166 mungbean genotypes for all the examined variables except for SPP and YMV (Table 6). The selection difference (SD) was positive for all variables, except for SPP and YMV. The heritability (h^2^) ranged from 0.81 for YLD to 0.997 for PPP (Table 6). Moreover, the selection gain (SG) was positive for all studied parameters except for SPP and YMV. The largest positive SG was 17.1% for YPP, even as DM had the lowest SG value of 1.84%, while the negative SG ranged from −3.03% for YMV to −1.61% for SPP.

## 3. Discussion

Morphological characterization has been crucial in determining the genetic diversity of the mungbean. For the efficient evaluation, maintenance, and use of genotypes, the level of genetic diversity must be investigated [28]. Accurate genotypic descriptions and organization of genetic diversity would help to determine breeding strategies and facilitate appropriate choices for germplasm conservation. We have characterized the 166 mungbean genotypes using 19 morphological characteristics as per the standard list of descriptors for mungbean by IBPGR-Biodiversity.

The main objective of mungbean breeding programs around the world is to breed for high production potential, preferred grain quality, and resistance to abiotic and biotic stresses. These aims can only be fulfilled when there is significant genetic variation within the germplasm available to the breeders. To achieve the breeding aims, breeders regularly exchange germplasm locally and globally. Investigating the level of genetic diversity is vital for the proper evaluation, management, and exploitation of germplasm [28]. As the breeding program mostly depends upon the degree of genetic diversity, morphological characterization is regarded as an essential step in the description and categorization of crop genetic resources [29]. Screening for qualitative features is essential to define the plant, and has become vital for crop registration and certification [30]. The plant descriptors are not only affected by local consumers’ choices and their socio-economic conditions but also have a significant impact on natural selection and evolution [31]. A noteworthy difference among traits was displayed for Analysis of variance which was corroborative to previous workers [32,33,34,35,36,37]. In the present investigation, the presence of a significant qualitative variance was observed for all the studied characteristics, supported by Tripathi et al. [38]. Qualitative morphological traits are known to have a significant effect on the development of diversity through natural or artificial human involvement. Consumer demand is influenced by factors such as seed shape, seed surface colour, and shine. Therefore, mungbean breeding efforts are controlled by local or regional selection. For example, the small-seeded mungbean is highly valued and priced above the bold-seeded type of mungbean in the northwestern parts of Bangladesh. Similarly, cultivars with green hypocotyls are selected over those with purple ones by bean sprouting firms [39]. Shiny green seed coat colour genotypes are generally preferred over those with dull seed coats. For example, the density and length of trichomes are known to alter the choice of insect pests of certain species [40]. In the Shannon–Weaver Diversity Indices (SDI), H’ was estimated to assess the diversity in qualitative characters at both vegetative and reproductive stages of the accessions (Table 3). Intermediate phenotypic diversity was exhibited in hypocotyl colour, stem colour, leaf colour, leafiness, terminal leaflet shape, and calyx colour (H’ = 0.50–0.75), and the rest of the traits indicated high phenotypic diversity (H’ ≥ 0.75) [41].

Descriptive statistics for the quantitative variables demonstrated the large genetic variation and similar variance was verified by Tahir et al. [42] and Kanavi et al. [43]. Moderate to high genetic differences were reported for phenological parameters such as DFF, DF, DM, PH, SV, HSW, and BYPP. Less variance among the accessions was identified in CPP, SPWt, ShWtPP, SWtPP, and YMV. A similar result was revealed by Azam et al. [11]. The estimated high CV observed in our research shows the large size of heterogeneity confirmed by Tripathi et al. [38] and Win et al. [44]. The variability of flowering time was repeated by Kanavi et al. [33] from 33 to 53 days, and Win et al. [34] from 31 to 75 days. In our investigation, the genotypes revealed earlier flowering lines, with early flowering assuring early maturity. PH has a significant variation, which was consistent with Muthuswamy et al. [45].

Skewness and kurtosis will help in evaluating relative mean performance and the form of the distribution of characteristics. Studies on distribution characteristics, such as skewness and kurtosis, provide information on the distribution pattern of the variables under consideration in the population. Skewness is the degree of deviation from the symmetrical bell curve or the normal distribution. It examines the absence of symmetry in the data distribution. If the skewness is between −0.5 and 0.5, the data are fairly distributed and symmetrical. Kurtosis is all about the distribution of tails, including flatness. Lower kurtosis levels in data collection are an indicator that the data have short tails or a lack of outliers. If the kurtosis is near 0, then a normal distribution is frequently assumed. These are termed mesokurtic distributions. If the kurtosis is smaller than zero, then the distribution is light-tailed and is termed a platykurtic distribution. If the kurtosis is larger than zero, then the distribution has heavier tails and is termed a leptokurtic distribution. Positive kurtosis denotes a relatively peaked distribution. Negative kurtosis reveals a relatively flat distribution. As with skewness, if the amount of kurtosis is too big or too little, there is a worry about the normality of the distribution [46]. In the current research, skewness revealed that the distribution of the population was usually skewed for the maximum of the features. All the attributes displayed the leptokurtic curve.

Shannon’s diversity index (H) is also another index that is generally used to characterize the species variety in a specific community. Shannon’s diversity index accounts for both the richness and evenness present in the species and is also used for a broad diversity of areas. The estimated H’ index ranged from 4.97 for YPP to 5.11 for DF, DM, PH, and SPP among the phenotypic features, but H’ index normally ranges from 1.5 to 3.5, but in rare cases can reach 4.5 [47]. Olukolu et al. [48] provided an H’ index of nineteen qualitative features (0.1 to 0.15) and twenty-eight numerical traits (0.09 to 0.16) of Bambara groundnut that corroborated our results. Bonny et al. [49] assessed the variety of qualitative features of Bambara groundnut landraces comparable to our result. These morphological differences in the genotypes indicate a possibility for development through breeding the crop and an urge for germplasm conservation [50].

The analysis of variance found that the tested genotypes were highly variable, and suggested that these were nearly all quantitative features. This is a chance to continue further breeding efforts to improve the characteristics of interest. From the pooled quantitative data of the two years, a significant variation was identified among the 19 various morpho-physiological features. The previous study also discovered that a wide range of genetic variability of economically significant features is in confirmation with the results of the current research [51,52]. These results conformed with the findings of Azam et al. [11]. Similar substantial variations in the yield and other aspects were observed by Kanavi et al. [43].

Features such as early flowering, synchronous maturity (flowering period), early maturity, pod length, no. of seeds per pod, and seed weight are the traits that have direct utilization in crop development programs. The development of short-duration cultivars may help in mungbean production in rice–wheat crop-rotation-based agricultural systems throughout the spring/summer season [53]. Change in rainfall patterns, including delayed monsoon, early termination, and insufficient and unequal distribution, has become a common phenomenon because of significant climate change [54]. In such a short-duration development environment, mungbean varieties would be extremely crucial in sustaining mungbean production. The maximum yield was found for genotype G12; however, accession G119 produced the lowest yield. Often, the best yield was achieved from the early maturity genotype (29.77 g), while the yield (1.9 g) of the late maturity genotypes was generally low. The results of this experiment on variance in studied properties of mungbean genotypes are in accordance with the earlier findings of Kindeya et al. [11,52]. This finding is in accord with the study of Sen and De [55], that indicated highly substantial variances among the mungbean genotypes for all the investigated 13 variables and thus supported the existence of large levels of diversity among the genotypes.

The principal component analysis is intended to produce a limited number of linear relationships of a set of variables that maintain much of the existing information in the original variables [56]. The PCA is one of a series of techniques for accumulating high-dimensional information and exploiting the dependency between the factors in a more faithful form without losing information. It indicates the major contribution to the overall difference in each differentiation axis. The PCA indicated that the gross variability found among the 166 genotypes can be described with six main components with eigenvalues larger than unity. The PCA generally confirmed the categorization achieved by cluster analysis, however it did not produce stable groupings. The first four principal components (PCs) explained roughly 64.60% of the total variance with individual contributions as 29.90%, 15.70%, 11.20%, and 7.96% accordingly. PCA is important for breeders to implement particular breeding strategies according to deep knowledge about the groups where certain traits are more important. The results of the study are more associated with the characterization of germplasm [57,58,59,60] that described three, four, two, three, five, three, four, three, and five principal components with 78.06%, 75.48%, 69.11%, 88.4%, 84.04%, 79.00%, 73.22%, 71.11%, and 76.00% of the total variation in mungbean, respectively. Furthermore, PCA was also employed for constructing biplots and investigating the relationship between genotypes and their agronomic characteristics. It may be revealed that there exists significant genetic diversity among the genotypes based on the distribution pattern of the genotypes on the biplot. Similar results for grouping a large number of green grams among genotypes by a variable biplot have been published by Kanavi et al. [43].

This is a statistical technique that clusters the sample values into groups based on the strong similarities found in the data set [61]. Ward [62] introduced agglomerative hierarchical cluster analysis in which a squared Euclidean distance is used for discovering similarities in the data set. Many research findings were utilized for the estimation of an optimal number of clusters [63].

Geographic isolation or genetic obstacles to ability crossing were caused by genetic divergence. It is essential to be aware of the level and pattern of genetic diversity within and between populations to find useful materials for plant breeding and to better understand the crop to design adequate collection and conservation procedures (accessions). For the creation of new varieties and to enhance output, it is essential to keep collecting and using genetically varied mungbean germplasms to strengthen the genetic base of parental lines. Besides the conventional breeding techniques, the use of wide hybridization to exploit wild species germplasm and to utilize the available genetic diversity of Gene Bank repositories characterized by high throughput genomic tools would constitute the right method.

Cluster analysis is a statistical technique for grouping items into clusters and figuring out how closely linked they are to one another. In the present study, the agglomerative hierarchical clustering method was employed in a cluster analysis for 166 genotypes using 18 morphological factors. The analysis separated the genotypes into seven categories. I intend to select seven genotypes from each cluster for diallel crossing using hybridization techniques. It was more difficult to cross if we had more than seven people packed together in the materials. Due to these factors, I created seven clusters out of the 166 mungbean genotypes. The clusters could be useful for heterotic breeding in the future since different sets of alleles may have an impact on a trait’s performance.

The clustering of a large set of genotypes into subsets of homogeneous clusters assists in the identification of various parents or genotypes for use in breeding or any other research programs. It aids in bringing the highest number of desired genes/traits into progenies with minimal effort [64]. It is also efficient in identifying accessions with relevant features belonging to distinct clusters for hybridization. The 166 accessions from various countries in the experiment were categorized into seven major clusters with a similarity coefficient of 0.409 based on morphological features, which is an indicator of genetic heterogeneity among the accessions. The clustering of genotypes was not correlated with the geographical origin of the genotypes, instead it shows the morphological similarity between them. Thus, geographic origin cannot be treated as the sole criterion for the identification of suitable donors for a breeding program. The genotypes from the same cluster group displayed similarities in their performance and were considerably different from other cluster groups. This suggests that crossing between outstanding genotypes of the abovementioned variety of varied cluster couples could produce appropriate recombinants for developing high-yielding mungbean cultivars. The findings of this study were quite related to those of many investigations, such as Tahir et al. [42], who described five clusters of 254 genotypes in mungbean; seven clusters of 196 genotypes in mungbean were described by Win et al. [44], seven clusters of 84 genotypes in mungbean were described by Sarkar and Kundagrami [28], and eleven clusters of 80 genotypes in mungbean were described by Sen and De [55]. These data reveal the significant number of genetic variations among the examined mungbean accessions and provides an excellent chance for the selection of parents.

The inter-cluster distances were higher than intra-cluster distances, which demonstrated the presence of a significant degree of genetic diversity across the studied genotypes. The higher the degree of intra- and inter-cluster distance, the higher the diversity among the clusters and within the cluster and vice versa. The degree of intra-cluster distances shows the magnitude of genetic diversity among genotypes in the same cluster. In the current research, the intra-cluster D^2^ values varied from 3.60 (cluster VII) to 5.15 (cluster II) (cluster II). The inter-cluster distance was larger than the intra-cluster distance, showing significant genetic variation among the genotypes. The results are in conformity with the findings of Sen and De [55].

Improvement in yield and other relevant traits is the basic objective in every breeding program. So, cluster diversity for seed production and its contributing features has to be evaluated for the selection of genotypes. In the present study, large differences were identified among the clusters for the majority of the characteristics investigated. Cluster III had the highest mean value for PPP (31.47), CPP (5.06), SPP (12.59), PWt (1.00), SWtPP (0.63), PL (8.65), HSW (44.86), YMV (2.64), YPP (20.99), and HI (58.24) (58.24). These results were closely linked with other studies [50,65,66]. Therefore, the data of this finding will be highly important, with some interesting characteristics for future plant breeding programs and the development of a new variety. Therefore, these clusters may be considered superior for choosing genotypes with desired features. Similar results were reported by Mehandi et al. [67] and Sarkar and Kundagrami [28].

As yield is the consequence of the combined effect of numerous component characteristics and environment, studying the interaction of traits among themselves and with the environment has been of great use in plant breeding. The correlation matrix is a popular technique for the evaluation of the degree of the relationship between two or more variables. For superior genotypes, selecting a program based on a consideration of the correlation matrix may be a great method of measurement [6]. Correlation tests demonstrated that out of the 19 morphological and agronomic parameters only PPP, CPP, SPP, SPWt, PL, and HSW showed a significant positive correlation with YPP. The result was in agreement with Parihar et al. [68]. These results are in agreement with those of Sheetal et al. [69] for the number of pods per cluster, while Hemavathy et al. [70] quoted similar results for the number of clusters per plant [61] and reported the same results for the number of pods per plant having a positive correlation with single plant yield. Ramachandra and Lavanya [71] observed a significant correlation between days to 50% blooming and days to maturity.

Plant breeders typically aim to integrate several suitable agronomic traits in one excellent genotype that finally leads to reaching excellent performance. In this context, several multivariate approaches are commonly utilized, such as principal component analysis, factor analysis, cluster analysis, and different samples to group measured traits or choose test genotypes [72]. In this context, Olivoto and Nardino [26] have developed a selection index for identifying genotypes and/or proposing treatments based on information on multiple traits. The multi-trait genotype–ideotype distance index (MGIDI) offers a more efficient and accurate treatment suggestion based on desirable or undesirable features for the crop investigated [73]. The multi-trait index based on factor analysis and ideotype design suggested the 33 accessions demonstrated the greatest performance and anticipated balanced and desirable genetic improvements for all the investigated variables. The accessions identified using the MGIDI index have the ability to improve all the attributes simultaneously. According to Olivoto et al. [74], the MGIDI offers a novel selection process that evaluates the correlation structure among the features. Therefore, based on all trait index strategies, the germplasms G45, G5, G22, G55, G143, G144, G87, G138, G110, G133, and G120 might be considered as the best parents based on the qualitative and quantitative characters, especially maximum yield per plant with high PL, PPP, and HSW.

## 4. Materials and Methods

### 4.1. Plant Materials

The mungbean germplasms (#166) utilized in the study were collected and generated from varied sources (Appendix A)—containing local landraces (#39), PRC advanced lines (#39), approved and released varieties (#8), and exotic excellent lines (#80). The advanced lines were collected through the Breeding Division at the Pulses Research Centre (PRC) of Bangladesh Agricultural Research Institute (BARI), and the released varieties from separate national institutions. The native landraces’ seeds were obtained randomly from diverse agroecosystems in Bangladesh. The exotic lines were collected from the Asian Vegetable Research and Development Center (AVRDC, Hyderabad, India); Australian Centre for International Agricultural Research (ACIAR, Canberra, ACT, Australia); Indian Institute of Pulses Research, (IIPR, Kanpur, India), SAARC Agriculture Centre (SAC, Dhaka, Bangladesh), two lines from Thailand, and two from China.

### 4.2. Experimental Field Conditions

The field experiment was conducted at the PRC of Bangladesh Agricultural Research Institute (BARI) research area located in Ishurdi (24.03° N 89.05° E, 16 m above mean sea level; Pabna, Bangladesh). The soil of the research field was assessed before seed sowing, and it was highland classified by sandy loam to clay loam in texture with extremely poor nutrient status. The physical and chemical parameters of the soil are presented in detail in Appendix A [75,76,77,78,79,80]. All the cultivars were sown in the 2018 Kharif II period or Season 1 (September to December), and a similar set of trials were seeded in the 2019 Kharif II period or Season 2 in the same month. Temperature, relative humidity (RH %), rainfall (mm), and photoperiod for the experimental location were obtained at the PRC meteorological site (Appendix A).

### 4.3. Experimental Design

The field experiment was carried out repetitively during the growth periods of two seasons (2018 and 2019). The trials were conducted in an open area and irrigated initially before cropping for seedling emergence. The field design was in a randomized complete block design (RCBD) with three replications. Seeds of mungbean were carefully sown in continuous seeding to a depth of 2–2.5 cm with a spacing of 30 cm in a full plot size of 152 m × 16 m. The spacing between two replicative blocks was kept at 1 m. The unit plot size was 2.4 m^2^ (4 m × 0.6 m), the spacing between the individual plants was kept at 6–8 cm, and between every row was 4 m, making adjustments to a total of 45–50 plants in a row and 100–110 plants in a plot from a continuous sowing plot.

### 4.4. Crops Management

The field was prepared for planting by three-fold harrowing and levelling. The experiment site was medium-high land and has a well-drainage system. In the plots, the soil was originally ploughed thoroughly and fertilized according to the instructions of BARI [81]. The first manually weeding was performed 2–3 weeks after seeding and was performed again before flowering. The seed was treated with fungicides carboxin 17.5% + thiram 17.5% FF (3 g kg^−1^ seed) before sowing for prevention against seed and soil-borne diseases. Aphid and thrips invasion were controlled by using dimethode (2 mL L^−1^ of water) at 30 days after sowing (DAS) and imidacloprid (0.5 mL L^−1^ of water) at 20 DAS. Such treatments were applied thrice at 7 days intervals.

### 4.5. Field Data Collection

Fifteen distinct qualitative and nineteen quantitative traits were assessed using descriptors specified for mungbean published by the International Board for Plant Genetic Resources [82], World Vegetable Center (WVC), and little modified descriptors provided by Bisht et al. [83]. Plots were screened from time to time and the qualitative and quantitative data were recorded on exact dates (Appendix A). For each trait, measurements were recorded from five randomly selected plants for each genotype per replicate. The qualitative parameters, such as hypocotyl colour, seedling vigour, stem colour, and leaf shape, were visually recorded at seedling and growth stages in the field condition. Stem colour, stem pubescence, leaf pubescence, leaf colour, leafiness, terminal leaflet shape, calyx colour, and corolla colour were visually recorded throughout the flowering stage. Pod beak shape, mature pod colour, pod curvature, and seed properties were taken during crop maturity, determined when 80% of the pods became stained a blackish coloration. Chlorophyll content was measured using the “Konica Minolta SPAD-502” by taking three averages of five flag leaves per plant according to Babar et al. [84]. Canopy temperature (°C) was calculated from each plant that used a Handheld Infrared Thermometer (Model AG-42, Tela Temp Corporation, Fullerton, CA, USA) on clean and vivid sunny days between 12:00 and 14:00, with a significant variation of the half to one meter from the top portion of the pot and about 50 (± 5) cm above the canopy with a relatively accurate angle of 30°−60° from straight, offering a canopy view of 10 cm × 25 cm. Observations were recorded on five randomly selected plants per replication for quantitative traits, namely, days to flowering (DF) were recorded when 50% of the plants were at the flowering stage, and maturity (DM) was recorded when 90% of the plants attained maturity, plant height (cm) (PH) was measured at maturity stage, as well as number of pods per plant (NPPP), number of pods per cluster (NPPC), number of seeds per pod (NSPP), pod weight with seed (PWt), shell weight per pod (ShWtPP), seed weight per pod (SWtPP), pod length (PL), 100-seed weight (HSW), yellow mosaic virus (YMV), yield per plant (YPP), stover yield per plant (SYPP), biological yield per plant (BYPP), and harvest index (HI) (Appendix A).

### 4.6. Statistical Analysis

We averaged each treatment from all the sample data of a trait to obtain a replication mean [85,86]. The average data of various traits were analysed statistically [87] and biometrically [88]. We used Statistix 8 software to obtain an analysis of variance (ANOVA) [89,90]. Optimum numbers of cluster and scree plots were calculated by Nbclust [91] and factoextra using R 4.1.3 [92] by different packages, respectively. The elbow technique from the Nbclust package was used to discover the appropriate number of clusters. Principal components analysis (PCA), eigenvalues, eigenvectors, and biplot analysis were conducted with factoextra [93] and GGbiplot [94], respectively, using R 4.1.3 software [92]. These PCAs were obtained from the correlation matrix. Data were adjusted to determine the genetic distance matrix using the Euclidean method distance and then hierarchical clustering was performed using Ward’s method [95]. For cluster analysis, algorithm hierarchical clustering (AHC) across 166 genotypes was detected using the statistical package cluster [96] and factoextra [93]. Heatmap was produced using Euclidean distance and average methods using pheatmap [97] and R functions chart. The correlation was assessed using Performance Analytics Peterson [98], and the default Pearson’s method was utilized to represent trait associations. To calculate the multi-trait stability index (MTSI), the below given equation was used [74]:$$\mathrm{MTSI}i_{\;}={\left[\sum_{j=1}^{f}{\left(F_{ij}-F_{j}\right)}^{2}\right]}^{0.5}$$
where MTSI is the multi-trait stability index for the *i*_th_ genotype, *F_ij_* is the *j*_th_ score of the *i*_th_ genotype, and *F_j_* is the *j*_th_ score of the ideotype. The genotype with the lowest MTSI is, accordingly, closest to the ideotype and therefore has a high mean performance and stability for all parameters assessed. The stability analysis of the multi-environment trial data using MTSI indices was performed using the metan package [74] in R 4.1.3 platform.

## 5. Conclusions

The results of various multivariate approaches can contribute to enhancing mungbean production. Most of the qualitative traits varied significantly across the germplasm. The highest diversity was identified qualitatively in seed colour, seedling vigour, leaf pubescence, stem pubescence, pod colour, and seed size. Descriptive statistics confirmed a higher degree of divergence among tested landraces. The analysis of variance indicated highly significant variations among the 166 genotypes for all the traits. Based on the data obtained from various multivariate approaches, all agronomic variables showed significant divergence. PPP, PPC, SPP, SPWt, PP, and HSW displayed strong relationships with YPP, which should be prioritized in breeding efforts to improve seed production. In PCA, ten components had a significant effect on total diversity, while cluster analysis helped in the selection of better genotypes for further use in the breeding effort. Based on morphological traits, 166 accessions from five countries were grouped into seven clusters with a similarity value of 0.409, with cluster II having the most genotypes (34) and cluster VII having the fewest (14). Depending on Mahalanobis D^2^ statistics, cluster II had the largest intra-cluster distance. Clusters II and VI (9.53) and I and VII (4.88) exhibited the largest and least inter-cluster distances, respectively. Considering all multivariate approaches for both qualitative and quantitative features, G45, G5, G22, G55, G143, G144, G87, G138, G110, G133, and G120 may be selected as optimal parents for the future breeding program. This information may help breeders to take initiative to design schemes for germplasm usage in the development and improvement of mungbean genotypes in the near future.

## Figures and Tables

**Figure 1 plants-12-01984-f001:**
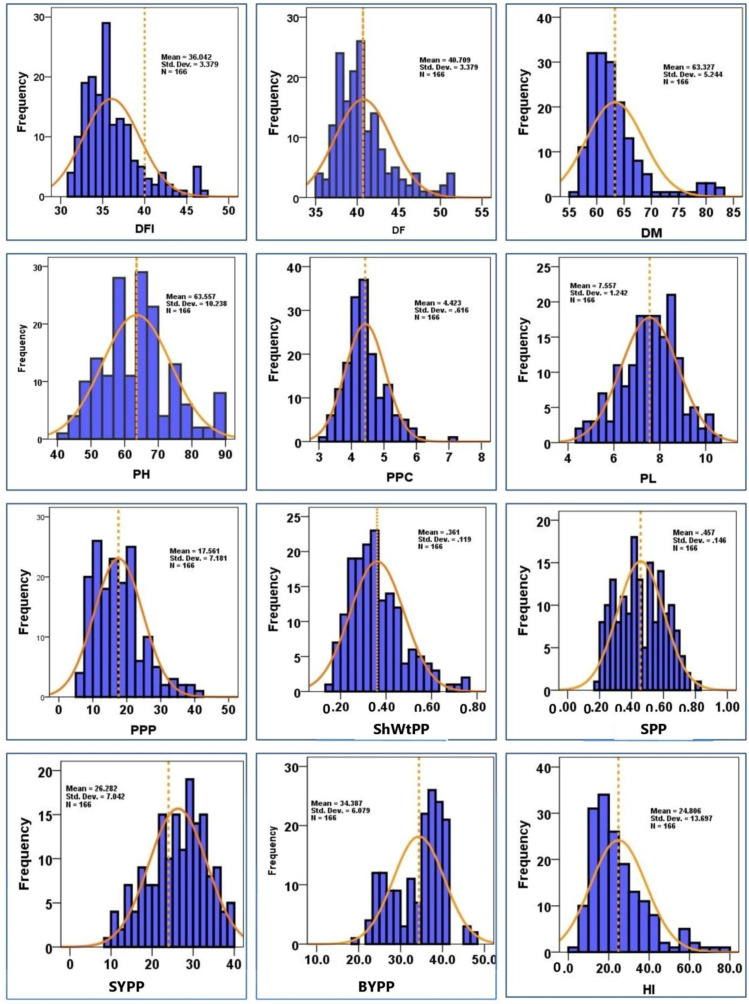
Histograms showing the frequency distribution curve of DFF = days to first flower, DF = days to flowering, DM = days to maturity, PH = plant height, CT = canopy temperature, SV = SPAD value, PPP = number of pods per plant, PPC = number of pods per cluster, SPP = number of seeds per pod, SPWt = pods weight with seed, ShWtPP = shell weight per pod, SWtPP = seeds weight per pod, PL = pod length, HSW = 100 seed weight, YMV = yellow mosaic virus, YPP = yield per plant, SYPP = stover yield per plant, BYPP = biological yield per plant, and HI = harvest index used as experimental attributes, respectively.

**Figure 2 plants-12-01984-f002:**
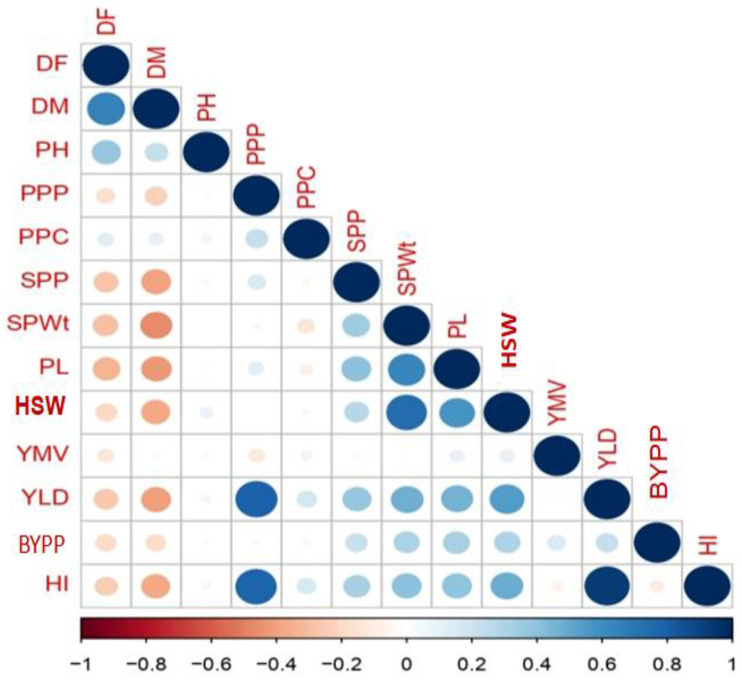
Correlation coefficients of grain yield on growth traits and yield components in *V. radiata.* According to the colour scale, a positive correlation is shown in blue and negative correlations are shown in red colour. The colour intensiveness and the size of the circle are relatively proportional to the correlation. Large and blue circles denote strong relationships and smaller circles denote weaker relationships. The colour scale indicates the extent of correlation, where 1 denotes completely positive relationships in dark blue and −1 denotes a completely negative correlation in dark red between two traits.

**Figure 3 plants-12-01984-f003:**
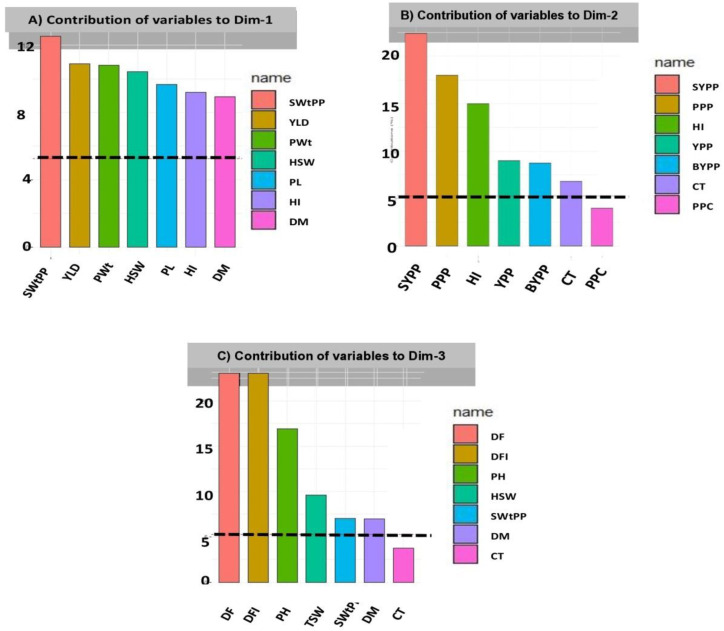
Contribution of variables to PCs (%) (**A**) PC1, (**B**) PC2, and (**C**) PC3 from the principal component analysis. Black dashed lines across bar plots are the reference lines and the variable bars above the reference lines are considered important in contributing to the respected PCs.

**Figure 4 plants-12-01984-f004:**
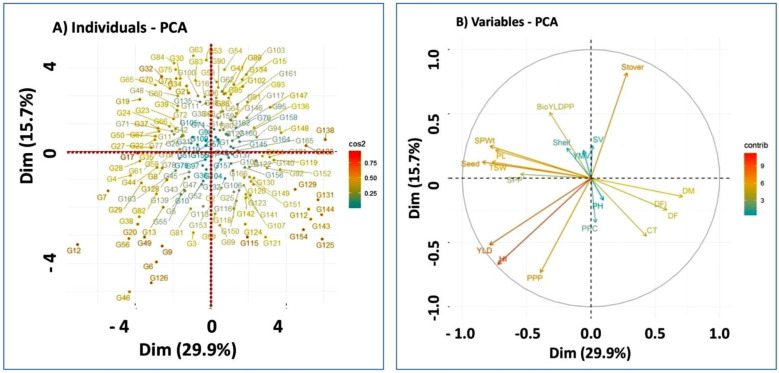
Graphical illustration of principal component analysis (PCA) score plots (**A**) of genotype vector of the distribution to each other based on the first two PCs; (**B**) of PCA on different variables in the first two principal components of mungbean genotypes. DFF = days to first flower, DF = days to flowering, DM = days to maturity, PH = plant height, CT = canopy temperature, SV = SPAD value, PPP = number of pods per plant, PPC = number of pods per cluster, SPP = number of seeds per pod, PWt = pods weight, ShWtPP = shell weight per pod, SWtPP = seeds weight per pod, PL = pod length, HHSW = 100 seed weight, YMV = yellow mosaic virus, YPP = yield per plant, SYPP = stover yield per plant, BYPP = biological yield per plant, and HI = harvest index used as experimental attributes, respectively.

**Figure 5 plants-12-01984-f005:**
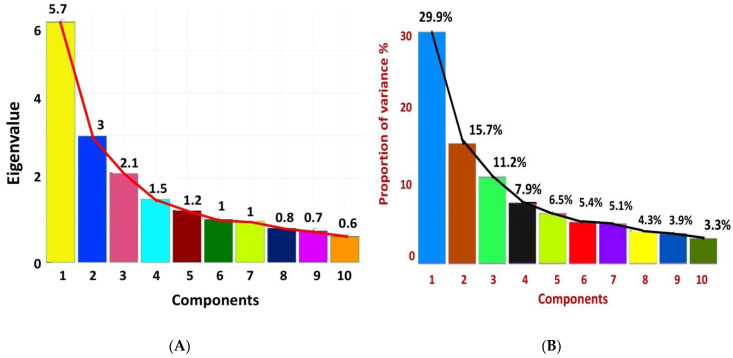
Scree plot showing (**A**) eigenvalue of first 10 PCs; (**B**) percent of variation explained by each principal component. The cumulative variation of 1 to 10 principal components is shown, whereas the black line explains variance in the phenotypic diversity of mungbean genotypes by each component.

**Figure 6 plants-12-01984-f006:**
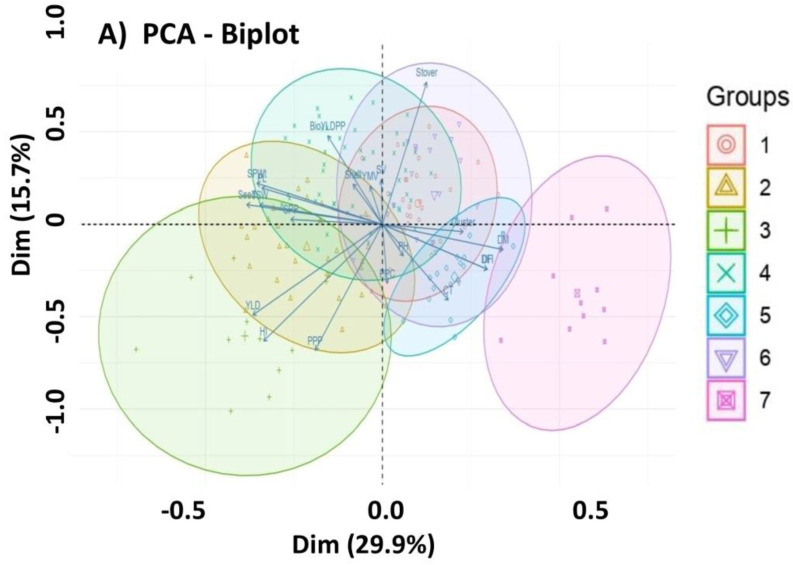
Genotype by trait: PCA ellipse biplot demonstrates clusters of mungbean accessions categorized by studied traits between PC1 and PC2 for 166 genotypes. Biplot analysis for phenotypic similarity. Correlated phenotypic components and genotype samples were located in the same quadrant. (**A**) Different symbols and colours are assigned to display different cluster groupings and (**B**) each ellipse represents accessions with specific yield-related traits, coloured ellipses highlight the observations taken in the different positions in the alley (the codes for genotypes and traits as described in Appendix A and Appendix A, respectively). The biplot shows the PCA scores of the explanatory variables as vectors in the sky in (**A**) black and (**B**) as individuals (i.e., colour marks) for each cluster. Individuals on the same side as a given variable should be interpreted as having a high contribution to it. The magnitude of the vectors (lines) shows the strength of their contribution to each PC. Vectors pointing in similar directions indicate positively correlated variables, vectors pointing in opposite directions indicate negatively correlated variables, and vectors at proximately right angles indicate low or no correlation. Coloured concentration ellipses (size determined by a 0.95 probability level) show the observations grouped by mark class.

**Figure 7 plants-12-01984-f007:**
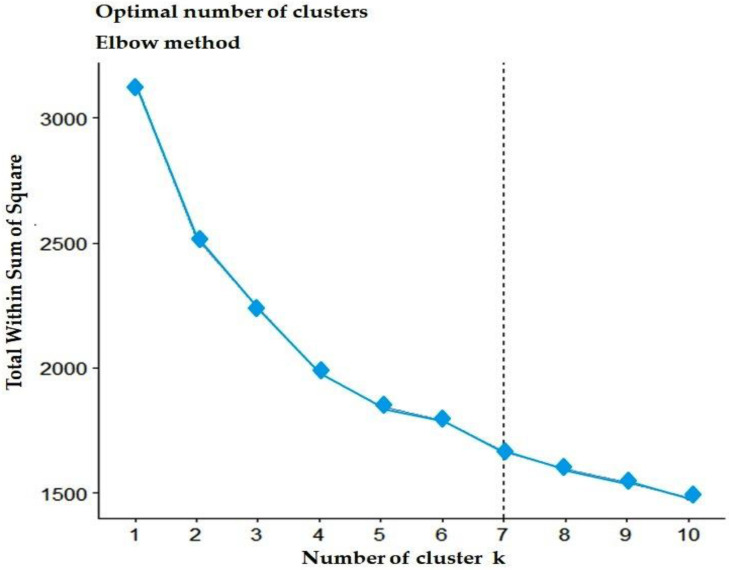
Determination of the optimal number of clusters using the elbow method based on quantitative traits. The best “k” is chosen at the point where the marginal gain sharply decreases, yielding an angle in the graph (the “elbow” criterion); in our case, the value k = 7 is optimal.

**Figure 8 plants-12-01984-f008:**
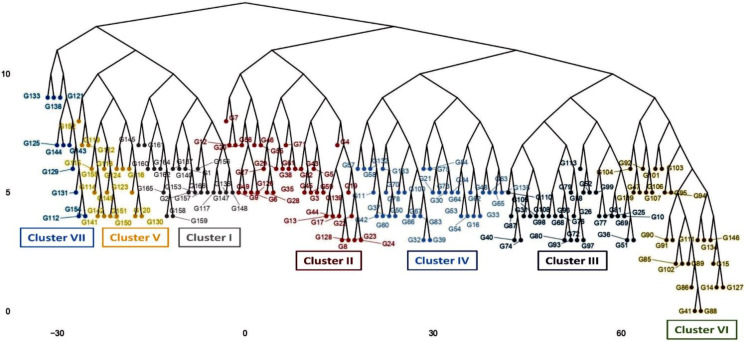
Clustering pattern (pooled over two seasons) of the 166 genotypes based on 19 morphological traits using Euclidean distance and ward clustering at a dissimilarity coefficient of 0.409. The name, code, and origin of genotypes are given in Table 2. Cluster I (grey), cluster II (pink), cluster III (violet), cluster IV (ash), cluster V (yellow), cluster VI (light green), and cluster VII (light blue).

**Figure 9 plants-12-01984-f009:**
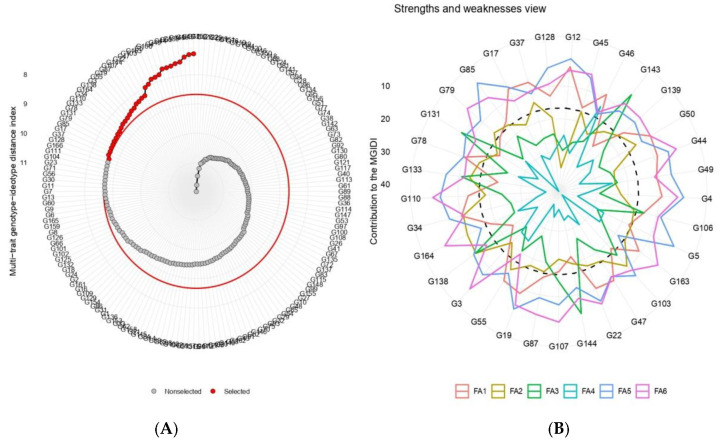
Genotype ranking (**A**) and selected genotypes (**B**) for the multi-trait stability index (MTSI) of 166 genotypes based on 19 traits. The selected genotypes are shown in red colour and the red circle represents the cut-point according to the selection differential of 11%.

**Table 1 plants-12-01984-t001:** Qualitative traits, descriptor states, frequency %, and Shannon–Weaver diversity index (H’) of mungbean genotypes.

Qualitative Traits	Descriptors States	Frequency %	H’
**Hypocotyl colour**	1 Green	26.51	0.625
	2 Green-purple	66.27	
	3 Purple	7.23	
	4 Dark purple	-	
	5 Mixed	-	
	6 Other	-	
**Seedling vigour**	3 Poor	27.11	1.076
	5 Intermediate	29.52	
	7 Vigorous	43.37	
**Stem colour**	1 Light green	44.58	0.687
	2 Dark green	55.42	
	3 Light purple	-	
	4 Dark purple	-	
	5 Others	-	
**Stem pubescence**	1 Glabrous	11.45	0.991
	2 Sparse	50.60	
	3 Moderately pubescent	37.35	
	4 Highly pubescent	0.60	
**Leaf pubescence**	1 Glabrous	19.28	1.042
	2 Very sparse	34.94	
	3 Sparsely pubescent	45.78	
	4 Moderately pubescent	-	
	5 Densely pubescent	-	
**Leaf colour**	3 Light green	22.89	0.735
	5 Intermediate green	37.95	
	7 Dark green	39.16	
**Leafiness**	1 Sparse	3.61	0.730
	2 Intermediate	22.89	
	3 Abundant	66.27	
**Terminal leaflet shape**	1 Deltate	70.48	0.592
	2 Ovate	24.70	
	3 Ovate-lanceolate	4.82	
	4 Lanceolate	-	
	5 Rhombic	-	
**Calyx colour**	1 Green	71.08	0.601
	2 Purplish-green	28.92	
	3 Greenish-purple	-	
	4 Others	-	
**Corolla colour**	1 Yellow	10.84	0.715
	2 Greenish yellow	49.40	
	3 Yellowish-green	39.76	
	4 Green-purplish yellow	-	
	5 Others	-	
**Pod beak shape**	1 Pointed	35.54	0.651
	2 Blunt/Round	64.46	
	3 Others		
**Mature pod colour**	1 Straw	4.22	0.927
	2 Coffee/chocolate		
	3 Brown	2.41	
	4 Brown and black	52.41	
	5 Black	40.96	
	6 Other	-	
**Pod curvature**	1 Straight	58.43	0.679
	2 Slightly curved	41.57	
	3 Curved (sickle-shaped)	-	
**Seed colour**	1 Light green	17.47	1.279
	2 Dark green	45.78	
	3 Light yellow	21.69	
	4 Yellow	15.06	
	5 Brown	-	
	6 Mottled	-	

**Table 2 plants-12-01984-t002:** Descriptive statistics and Shannon diversity index (SDI) for quantitative-yield-associated traits among 166 mungbean genotypes over two seasons.

Quantitative Traits	Range		Mean	SEM (±)	SD	Var	CV	Skewness	Kurtosis	SDI	Evenness (E_H_)
	Min	Max									
DFF	31.00	47	36	0.212	3.08	11.12	4.24	1.21	1.54	5.09	1.00
DF	36	51	41	0.262	3.38	11.42	33.76	1.29	1.64	5.11	1.00
DM	56.17	83	63	0.407	5.24	27.49	1.89	1.93	3.99	5.11	1.00
PH	41.74	89.77	63.56	0.795	10.24	104.81	1.22	0.45	0.07	5.11	0.99
CT	12.22	29.78	20.69	0.438	5.64	31.84	2.79	0.08	−1.37	5.08	0.96
SV	19.88	60.50	38.59	0.556	7.17	51.39	1.66	0.35	0.83	5.10	0.98
PPP	5	42	18	0.557	7.18	51.56	3.67	0.90	0.77	5.03	0.92
CPP	3	7	5	0.048	0.62	0.38	12.69	0.88	1.53	5.10	0.99
SPP	9	15	12	0.073	0.95	0.89	9.15	−0.18	0.65	5.11	1.00
SPWt	0.46	1.37	0.82	0.014	0.19	0.03	6.13	0.44	0.04	5.09	0.98
ShWtPP	0.16	0.76	0.36	0.009	0.12	0.01	17.28	0.98	1.00	5.06	0.95
SWtPP	0.20	0.80	0.46	0.011	0.15	0.02	8.82	0.12	−0.97	5.06	0.95
PL	4.53	10.55	7.56	0.096	1.24	1.54	1.49	−0.21	−0.19	5.10	0.99
HSW	17.81	56.44	35.40	0.773	9.96	99.15	4.69	0.21	−0.99	5.07	0.96
YPP	1.98	29.77	8.52	0.383	4.94	24.37	11.19	1.53	2.97	4.97	0.86
SYPP	9.14	40.00	26.28	0.547	7.04	49.58	4.81	−0.32	−0.60	5.07	0.96
BYPP	19.72	47.74	34.39	0.472	6.08	36.95	2.40	−0.45	−0.69	5.10	0.98
HI	4.90	76.15	24.81	1.063	13.70	187.62	10.81	1.30	1.74	4.98	0.87

SEM = standard error of the mean, SD = standard deviation, Var = variance, SDI = Shannon diversity index, DFF = days to first flower, DF = days to flowering, DM = days to maturity, PH = plant height, CT = canopy temperature, SV = SPAD value, PPP = number of pods per plant, PPC = number of pods per cluster, SPP = number of seeds per pod, SPWt = pods weight with seed, ShWtPP = shell weight per pod, SWtPP = seeds weight per pod, PL = pod length, HSW = 100 seed weight, YMV = yellow mosaic virus, YPP = yield per plant, SYPP = stover yield per plant, BYPP = biological yield per plant, and HI = harvest index.

**Table 3 plants-12-01984-t003:** Pooled analysis of variance for yield and different agro-morphological traits for the 166 mungbean genotypes studied over two years.

**Mean Squares**	**Traits**	**Sources of Variation**				
		**Replication within the Year (df = 4)**	**Genotype ** **(df = 165)**	**Year ** **(df = 1)**	**Genotype × Year** **(df = 165)**	**Pooled Error ** **(df = 660)**
	**DFF**	194.91	68.52 ***	76.48 ^ns^	11.04 ***	4.24
	**DF**	122.74	67.52 ***	76.48 ^ns^	10.04 ***	4.14
	**DM**	40.59	164.97 ***	820.50 **	5.74 ***	2.04
	**PH**	4.10	419.03 ***	898.72 ***	125.61 ***	35.26
	**CT**	8.03	191.02 ***	316.36 **	0.61 **	0.45
	**SPAD**	0.54	308.32 ***	1031.06 ***	1.33 ***	0.65
	**PPP**	126.11	309.39 ***	9740.14 ***	3.29 ***	0.52
	**CPP**	3.81	1.39 ***	51.04 **	0.99 ***	0.60
	**SPP**	59.37	5.36 ***	1552.34 *	4.19 ***	2.28
	**SPWt**	1.30	0.21 ***	13.01 *	0.07 *	0.12
	**ShWtPP**	0.93	0.09 ***	4.78 *	0.04 ^ns^	0.21
	**SWtPP**	0.07	0.13 ***	2.02 *	0.03 ^ns^	0.12
	**PL**	132.68	9.26 ***	99.22 ^ns^	0.04 **	0.13
	**HSW**	199.75	594.90 ***	9639.06 **	5.46 ***	3.73
	**YPP**	64.43	146.20 ***	2340.43 **	5.26 ***	1.47
	**SYPP**	45.45	287.94 ***	3233.28 **	3.01 ***	2.03
	**BYPP**	13.34	221.71 ***	11074.65 ***	2.55 ***	0.69
	**HI**	486.36	1125.70 ***	3855.41 *	6.28 ^ns^	10.80

DFF = days to first flower, DF = days to flowering, DM = days to maturity, PH = plant height, CT = canopy temperature, SV = SPAD value, PPP = number of pods per plant, PPC = number of pods per cluster, SPP = number of seeds per pod, SPWt = pods weight with seed, ShWtPP = shell weight per pod, SWtPP = seeds weight per pod, PL = pod length, HSW = 100 seed weight, YMV = yellow mosaic virus, YPP = yield per plant, SYPP = stover yield per plant, BYPP = biological yield per plant, and HI = harvest index. *, ** and *** = significant at 0.05, 0.01 and 0.001 level of probability, respectively, ns = non-significant.

**Table 4 plants-12-01984-t004:** Average intra-cluster (main diagonal) and inter-cluster (off-diagonal) of 166 genotypes using Mahalanobis D^2^ analysis.

Clusters	I	II	III	IV	V	VI	VII
**I**	**4.31**						
**II**	6.94	**5.15**					
**III**	5.13	5.69	**4.02**				
**IV**	6.05	6.12	5.25	**5.03**			
**V**	5.46	6.84	5.00	5.60	**4.74**		
**VI**	7.07	9.53	7.41	8.77	6.81	**3.74**	
**VII**	4.88	7.09	5.02	6.74	5.84	6.68	**3.60**

**Table 5 plants-12-01984-t005:** Mean values and standard deviation of quantitative variables in different clusters of mungbean genotypes.

Traits	Clusters						
	1	2	3	4	5	6	7
**DFF**	34.81 ± 1.42	34.54 ± 2.18	35.14 ± 2.14	34.74 ± 2.11	35.57 ± 1.61	40.11 ± 2.64	44.25 ± 2.07
**DF**	39.48 ± 1.42	39.2 ± 2.18	39.80 ± 2.14	39.40 ± 2.11	40.24 ± 1.61	44.77 ± 2.64	48.92 ± 2.07
**DM**	64.79 ± 3.72	60.05 ± 2.09	59.80 ± 1.28	61.65 ± 2.45	62.81 ± 2.44	64.72 ± 3.05	79.77 ± 2.02
**PH**	57.84 ± 7.83	66.46 ± 9.08	63.51 ± 8.09	62.71 ± 8.7	57.52 ± 7.57	67.07 ± 12.81	77.64 ± 9.25
**CT**	26.01 ± 2.25	18.85 ± 5.02	22.98 ± 4.32	16.33 ± 3.93	26.1 ± 4.53	17.52 ± 1.87	27.25 ± 1.64
**SPAD**	41.7 ± 8.35	38.86 ± 5.28	34.86 ± 6.66	38.29 ± 7.15	36.62 ± 5.41	40.4 ± 9.24	34.26 ± 2.25
**PPP**	17.11 ± 5.25	21.39 ± 6.01	31.47 ± 6.88	13.96 ± 4.66	20.23 ± 6.5	12.61 ± 4.37	13.94 ± 3.99
**CPP**	4.57 ± 0.5	4.25 ± 0.43	5.06 ± 1.10	4.38 ± 0.61	4.19 ± 0.51	4.31 ± 0.56	4.71 ± 0.51
**SPP**	11.62 ± 0.94	12.02 ± 0.76	12.59 ± 1.11	11.86 ± 0.78	11.24 ± 0.85	11.26 ± 0.9	10.48 ± 0.5
**SPWt**	0.67 ± 0.11	0.94 ± 0.14	1.00 ± 0.16	0.89 ± 0.17	0.63 ± 0.08	0.83 ± 0.1	0.59 ± 0.09
**ShWtPP**	0.32 ± 0.1	0.41 ± 0.11	0.36 ± 0.11	0.35 ± 0.15	0.37 ± 0.08	0.39 ± 0.12	0.31 ± 0.07
**SWtPP**	0.35 ± 0.07	0.54 ± 0.09	0.63 ± 0.09	0.53 ± 0.11	0.27 ± 0.03	0.44 ± 0.11	0.28 ± 0.1
**PL**	7.03 ± 1.11	8.25 ± 0.89	8.65 ± 1.15	7.88 ± 1.04	6.24 ± 0.55	7.93 ± 0.68	5.55 ± 0.74
**HSW**	27.34 ± 5.55	41.52 ± 6.88	44.86 ± 5.6	40.73 ± 8.07	22.19 ± 2.07	35.18 ± 6.48	25.36 ± 7.23
**YMV**	2.48 ± 1.11	2.05 ± 0.89	2.64 ± 1.26	2.55 ± 1.44	1.85 ± 0.82	1.66 ± 0.47	1.68 ± 0.54
**YPP**	6.17 ± 2.07	12.13 ± 3.23	20.99 ± 4.73	7.73 ± 2.68	5.74 ± 2.03	5.70 ± 2.55	3.95 ± 1.54
**SYPP**	31.31 ± 4.43	21.11 ± 5.54	15.02 ± 4.72	30.33 ± 3.59	19.56 ± 2.74	30.22 ± 5.08	25.30 ± 6.86
**BYPP**	36.98 ± 4.14	32.19 ± 6.86	35.78 ± 4.58	37.58 ± 3.31	25.00 ± 2.91	35.44 ± 4.81	28.85 ± 5.53
**HI**	16.68 ± 5.68	36.86 ± 8.53	58.24 ± 10.59	20.37 ± 6.6	22.67 ± 7.48	16.03 ± 6.64	14.67 ± 7.78

DFF = days to first flower, DF = days to flowering, DM = days to maturity, PH = plant height, CT = canopy temperature, SPAD = soil plant analysis development (SPAD), PPP = number of pods per plant, CPP = number of cluster per plant, SPP = number of seeds per pod, SPWt = single pod weight, ShWtPP= shell weight per pod, SWtPP = seed weight per pod, PL = pod length, HSW = 1000 seed weight, YMV = yellow mosaic virus, YPP = yield per plant (g), SYPP = stover yield per plant, BYPP = biological yield per plant (g), and HI = harvest index used as experimental attributes, respectively.

**Table 6 plants-12-01984-t006:** Estimates of selection differential, selection gain, and heritability based on MTSI for seven seed compositions evaluated for 166 mungbean germplasms.

Variables	Factor	Xo	Xs	SD	SD (%)	SG	SG (%)	h^2^
CT	FA1	19.7	20.2	0.459	2.33	0.451	2.29	0.984
PWt	FA1	0.842	0.89	0.0485	5.76	0.0478	5.68	0.986
SWtPP	FA1	0.477	0.506	0.029	6.08	0.0288	6.03	0.992
SPP	FA1	12	11.8	−0.198	−1.65	−0.193	−1.61	0.975
PL	FA1	7.4	7.57	0.164	2.22	0.159	2.15	0.969
HSW	FA1	3.7	3.98	0.271	7.33	0.269	7.27	0.992
SYPP	FA2	27.6	28.4	0.816	2.96	0.811	2.94	0.993
BYPP	FA2	35.4	38.2	2.73	7.7	2.61	7.35	0.955
DFF	FA3	36.9	38	1.18	3.19	1.09	2.96	0.927
DF	FA3	42.7	44.3	1.66	3.89	1.62	3.79	0.975
DM	FA3	65.6	66.8	1.23	1.87	1.21	1.84	0.982
PH	FA3	63.3	69.7	6.35	10	6.24	9.86	0.983
PPP	FA4	18.5	23.3	4.74	25.6	4.73	25.5	0.997
CPP	FA4	4.27	4.78	0.506	11.8	0.454	10.6	0.897
YPP	FA4	7.85	9.51	1.65	21.1	1.34	17.1	0.812
HI	FA4	22	25.1	3.06	13.9	2.84	12.9	0.93
ShWtPP	FA5	0.365	0.384	0.0194	5.32	0.0187	5.12	0.963
SV	FA6	38.2	39.3	1.04	2.73	1.03	2.69	0.987
YMV	FA6	2.17	2.11	−0.0698	−3.22	−0.0657	−3.03	0.941

Xo: overall mean of genotypes; Xs: mean of the selected genotypes; SD: selection differential; SG: selection gain or impact; h^2^: heritability. DFF = days to flower initiation, DF = days to flowering, DM = days to maturity, PH = plant height, CT = canopy temperature, SV = SPAD value, PPP = number of pods per plant, CPP = number of clusters per plant, SPP = number of seeds per pod, SPWt = single pod weight, ShWtPP = shell weight per pod, SWtPP = seed weight per pod, PL = pod length, HSW = 1000 seed weight, YMV = yellow mosaic virus, YPP = yield per plant (g), SYPP = stover yield per plant, BYPP = biological yield per plant(g), and HI = harvest index used as experimental attributes, respectively.

## Data Availability

Data recorded in the current study are available in all tables and figures of the manuscript.

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
