# Peer review of "Genetic Analyses of Mungbean [Vigna radiata (L.) Wilczek] Breeding Traits for Selecting Superior Genotype(s) Using Multivariate and Multi-Traits Indexing Approaches"

_plants, 2023, doi:10.3390/plants12101984_

Round 1

Reviewer 1 Report

I have completed the review of the article:

  Genetic analyses of mungbean [Vigna radiata (L.) Wilczek]  breeding traits for selecting superior genotype(s) using multi- variate and multi-traits indexing approaches.

 It is an interesting study in the context of the Mungbean breeding. The manuscript includes good results. The results are quite clearly presented. However, a few things need to be corrected or explained

1-      The abstract and conclusion sections are so long

2-      In the abstract, Don’t use abbreviation (PCA) without explanation

3-      Keywords: please do not repeat any words already mentioned in the title!

4-         Line 118  please delete “ Literature has shown a lot of qualitative studies straits for qualitative evaluation of variabilities, like minerals, proximate compositions, vitamins, pigments, flavonoid content, phenolics, and antioxidant activity [20-23] in amaranths” it is not suitable to use amaranths plants in the manuscript about mungbean. Moreover, these references for the author (Sarker) of the present manuscript  

5-      Line 380: please delete “Literature has also shown corroborative results in agronomic traits of Zea mays [33,34], 380 Oryza sativa [35,36], and agronomic traits of A. spp. [37,38].

6-      In the references section, please delete references from 33 to 38, these references for the authors (Azam and Sarker) of the present manuscript and about Maize and rice

7-      Reference [14] missed in the manuscript

8-      Line 551, reference [13] isn’t in its right place

9-      Line 687, please delete (14)

10-  Line 898, please change the font of the reference [85]

Author Response

Reviewer 1 (Round 1)

Comments: I have completed the review of the article: Genetic analyses of mungbean [Vignaradiata(L.) Wilczek] breeding traits for selecting superior genotype(s) using multi- variate and multi-traits indexing approaches.

 It is an interesting study in the context of the Mungbean breeding. The manuscript includes good results. The results are quite clearly presented. However, a few things need to be corrected or explained

Author response: Thank you for allowing us the opportunity to submit our revised manuscript for publication in Plants of MDPI. We appreciate the time and effort you have taken to improve our manuscript. We are also thankful to the honorable reviewer for the positive decision to publish in Plants of MDPI. We revised our manuscript following your point-by-point comments and suggestions for substantial improvement. We hope that this revised version satisfies you to take the final decision.

Comments: 1-The abstract and conclusion sections are so long

Author response: Thank you for your comments. We have concise it according to your suggestions.

Comments: 2-In the abstract, Don’t use abbreviation (PCA) without explanation

Author response: Thank you for your comments. We have defined the abbreviation of PCA.

Comments: 3-Keywords: please do not repeat any words already mentioned in the title!

Author response: Thank you for your comments. We have deleted the repeated words of the title.

Comments: 4- Line 118  please delete “ Literature has shown a lot of qualitative studies straits for qualitative evaluation of variabilities, like minerals, proximate compositions, vitamins, pigments, flavonoid content, phenolics, and antioxidant activity [20-23] in amaranths” it is not suitable to use amaranths plants in the manuscript about mungbean. Moreover, these references for the author (Sarker) of the present manuscript

Author response: Thank you for your comments. We have deleted it.

Comments: 5- Line 380: please delete “Literature has also shown corroborative results in agronomic traits of Zea mays [33,34], 380 Oryza sativa [35,36], and agronomic traits of A. spp. [37,38].

In the references section, please delete references from 33 to 38, these references for the authors (Azam and Sarker) of the present manuscript and about Maize and rice

Author response: Thank you for your comments. We have deleted it.

Comments: 7-Reference [14] missed in the manuscript

Author response: Thank you for your comments. We have corrected the reference number chronologically.

Comments: 8- Line 551, reference [13] isn’t in its right place

Author response: Thank you for your comments. We have deleted it.

Comments: 9- Line 687, please delete (14)

Author response: Thank you for your comments. We have deleted it.

Comments: 10- Line 898, please change the font of the reference [85]

Author response: Thank you for your comments. We have changed the font of the reference.

Reviewer 2 Report

Comments for manuscript “Genetic analyses of mungbean [Vigna radiata (L.) Wilczek] breeding traits for selecting superior genotype(s) using multi-variate and multi-traits indexing approaches”

The manuscript employed multi-variate and multi-traits indexing approaches for characterizing different mungbean genotypes and selecting superior genotypes with  15 qualitative traits of 166 diverse mungbean genotypes observed in two cultivated seasons. The main contribution of the manuscript was showing the diversity and the relationship among the genotypes by phenotype-based analysis with PCA and agglomerative hierarchical analysis, and then selected potential potential superior genotypes with multi-trait stability index (MTSI).  The authors had conducted a comprehensive data analyses for exploiting the results and conclusions. However, in my point of view, I am not convinced in several major points. 

major points:

1. For showing the scope of the analysis, the authors should show the representative of the 166 diverse mungbean genotypes employed in the study., eg, how these genotypes could represent all of the mungbean natural diversity (or all of the mungbean diversity in a specific breeding backbone group)

2. The authors should focus more on selecting the superior genotypes, rather than on describing the characterization of the clusters. eg. analyses using different weight on different traits for constructing the multiple-traits index for exploiting the superior genotypes.

3. For fig7, the elbow plot did not converge at k = 10. More k values should also be analyzed for a better choice of the k in case of convergence. 

4. The authors should also explain why all materials were clustered into 7 groups. In addition, the clustering results were not according to the origin of the genotypes might be limited by the phenotype data as these data were collected from two places. In general, the clustering results from the genotypic data are according to the geographic origin. Therefore, the authors should discuss more on why the clustering results are uncorrelated with the geographic origin.

Minor points:

1. The authors should describe the plant materials used in the study at the beginning of the results, eg, what materials collected from what places and representing what potential sub-groups or major groups, but not just in the plant materials section.

2. In line 210, the authors attempted to exploit the correlation but no the association among the traits, since only correlation analysis was conducted.

3. In line 303, the fig8 is the clustering tree, but no the phylogenetic tree which generally informed from genotypic data or genetic evidences. 

The abstract should be shortened by summarized the results into several major points.

Author Response

Reviewer 2 (Round 1)

Comments: Comments for manuscript “Genetic analyses of mungbean [Vigna radiata (L.) Wilczek] breeding traits for selecting superior genotype(s) using multi-variate and multi-traits indexing approaches”

The manuscript employed multi-variate and multi-traits indexing approaches for characterizing different mungbean genotypes and selecting superior genotypes with 15 qualitative traits of 166 diverse mungbean genotypes observed in two cultivated seasons. The main contribution of the manuscript was showing the diversity and the relationship among the genotypes by phenotype-based analysis with PCA and agglomerative hierarchical analysis, and then selected potential potential superior genotypes with multi-trait stability index (MTSI).  The authors had conducted a comprehensive data analyses for exploiting the results and conclusions. However, in my point of view, I am not convinced in several major points. 

Author response: Thank you for allowing us the opportunity to submit our revised manuscript for publication in Plants of MDPI. We appreciate the time and effort you have taken to improve our manuscript. We are also thankful to the honorable reviewer for the positive decision to publish in Plants of MDPI. We revised our manuscript following your point-by-point comments and suggestions for substantial improvement. We hope that this revised version satisfies you to take the final decision.

major points:

Comments: 1. For showing the scope of the analysis, the authors should show the representative of the 166 diverse mungbean genotypes employed in the study., eg, how these genotypes could represent all of the mungbean natural diversity (or all of the mungbean diversity in a specific breeding backbone group)

Author response: Thank you for your comments. The genotypes we used in this experiment were collected from home and abroad. This germplasm represents most of the mungbean growing country. Most of the materials were collected from the mini-core collection of the world vegetable center (WVC). In this study, some genotypes were collected from Thailand, Pakistan, India, and China. So, I think germplasm represents most of the natural diversity of the globe.

Comments: 2. The authors should focus more on selecting the superior genotypes, rather than on describing the characterization of the clusters. eg. analyses using different weight on different traits for constructing the multiple-traits index for exploiting the superior genotypes.

Author response: Thank you for your comments. We have rewritten some sentences (please see lines 346-355).

Comments: 3. For fig7, the elbow plot did not converge at k = 10. More k values should also be analyzed for a better choice of the k in case of convergence. 

Author response: Thank you for your comments. In cluster analysis, the elbow approach is a heuristic used to decide the number of clusters in a statistic set. A necessary step for any unsupervised algorithm is to decide the optimal quantity of clusters into which the data can be grouped. The elbow approach is one of the most famous strategies for finding out the top of the value of k. The elbow graph indicates the Within-Cluster Sum of Square (WCSS) values (on the y-axis) relative to specific K values (on the x-axis). In the elbow method, we are actually varying the number of clusters (K) from 1 to 10. For every value of K, we calculate the WCSS. WCSS is the sum of the square distances between every point and the centroid in a cluster. When we plot the WCSS versus the K value, the format appears like an elbow. As the number of clusters increases, the WCSS value begins to decrease. The value of WCSS is the highest when K = 1. When we analyze the graph, we can see that the graph changes at one point and then graph an elbow shape. From this point, the graph starts to move nearly parallel to the X-axis. The K value related to this point is the optimal K value or an optimum number of clusters. Beyond the elbow point, increasing the value of ‘K’ does not guide to a significant decrease in WCSS. For these reasons, we can choose 7 in the case of clustering.

Comments: 4. The authors should also explain why all materials were clustered into 7 groups. In addition, the clustering results were not according to the origin of the genotypes might be limited by the phenotype data as these data were collected from two places. In general, the clustering results from the genotypic data are according to the geographic origin. Therefore, the authors should discuss more on why the clustering results are uncorrelated with the geographic origin.

Author response: Thank you for your comments. For your kind consideration, clustering is an unsupervised machine learning method that can recognize groups of similar data points, called clusters, from data alone. There are more than thirty indicators and methods to identify the optimal number of clusters. So, we only focus on the common method to identify a good number of clusters. Arguably the most well-known method is the elbow method, in which the sum of squares is calculated and plotted for each number of clusters, and the user looks for a change of slope from steep to flat (elbow) to determine the optimal number of clusters. This method is imprecise, but still potentially useful. The elbow wind system is helpful because it shows how adding the number of clusters contributes to separating the clusters in a meaningful way, not in a borderline way. The Elbow method is fairly clear, if not a naïve solution based on intra-cluster variance. The gap statistic is a more sophisticated method to deal with data that has a distribution with no obvious clustering. This method, however, is somewhat subjective, as different people may identify the elbow at different locations. In our example in Figure 7, some may argue that k=7 is the elbow. From this measurement (Fig 7) it appears 7 clusters would be the appropriate choice.

Minor points:

Comments: 1. The authors should describe the plant materials used in the study at the beginning of the results, eg, what materials collected from what places and representing what potential sub-groups or major groups, but not just in the plant materials section.

Author response: Thank you for your comments. We have included some sentences (please see line 142-147).

Comments: 2. In line 210, the authors attempted to exploit the correlation but no the association among the traits, since only correlation analysis was conducted.

Author response: Thank you for your comments. We have changed the word “association” to “correlation”.

Comments: 3. In line 303, the fig8 is the clustering tree, but no the phylogenetic tree which generally informed from genotypic data or genetic evidences. 

Author response: Thank you for your comments. We have changed the words “polygenetic tree” to “clustering tree”.

Comments: The abstract should be shortened by summarized the results into several major points.

Author response: Thank you for your comments. We revised the abstract to shorten it by summarizing the results into several major points.

Round 2

Reviewer 2 Report

1. I would recommend that the authors discuss more on why the materials were clustered in to 7 groups. 

2. I would also recommend that the authors focus more on selecting the superior genotypes by testing different trait index strategies, which is generally interested by the readers in the field and many traits were investigated in the study.

 3. The quality of all figures should be improved for publication, the word size and word type should be consistent across all figures. 

Author Response

Response to the comments of Reviewer 2 (Round 2)

Comments: I have completed the review of the article: Genetic analyses of mungbean [Vigna radiata (L.) Wilczek] breeding traits for selecting superior genotype(s) using multi-variate and multi-traits indexing approaches.

Author response: Thank you once again for allowing us the opportunity to submit our re-revised manuscript for publication in Plants of MDPI. We appreciate the time and effort you have taken to improve our manuscript. We are also thankful to the honorable reviewer for the positive decision to publish in Plants of MDPI. We revised our manuscript following your point-by-point comments and suggestions for substantial improvement. We hope that this revised version satisfies you to take the final decision.

Comments:1- I would recommend that the authors discuss more on why the materials were clustered in to 7 groups. 

Author response: Thank you for your comments. We have added more in the discussion part according to your suggestions. We have written the following information clustered into different groups. (Lines 496-514)

“Geographic isolation or genetic obstacle to ability-crossing was caused by genetic divergence. It is essential to be aware of the level and pattern of genetic diversity within and between populations to find useful materials for plant breeding and to better understand the crop to design adequate collection and conservation procedures (accessions). For the creation of new varieties and to enhance output, it is essential to keep collecting and using genetically varied mungbean germplasm to strengthen the genetic base of parental lines. Besides the conventional breeding techniques, the use of wide-hybridization to exploit wild species germplasm and utilization of the available genetic diversity of Gene Bank repositories characterized by high throughput genomic tools would constitute the right method.

Cluster analysis is a statistical technique for grouping items into clusters and figuring out how closely linked they are to one another. In the present study, the agglomerative hierarchical clustering method was employed in a cluster analysis for 166 genotypes using 18 morphological factors. The analysis separated the genotypes into seven categories. I intend to select seven genotypes from each cluster for diallel crossing using hybridization techniques. It was more difficult to cross if we had more than seven people packed together in the materials. Due to these factors, I created seven clusters out of the 166 mungbean genotypes. The clusters could be useful for heterotic breeding in the future since different sets of alleles may have an impact on a trait's performance.”

Comments: 2- I would also recommend that the authors focus more on selecting the superior genotypes by testing different trait index strategies, which is generally interested by the readers in the field and many traits were investigated in the study.

Author response: Thank you for your nice comments. We have added these sentences in the manuscript based on clustering “Based on statistical analysis, the genotypes from cluster III might be considered as best parents for PPP, CPP, PL, HSW, and YPP as high yielding promising lines and would be used as distance parents for hybridization program” (Lines 340-343)

We also have added these sentences in the manuscript “Therefore, based on all trait index strategies the germplasm G45, G5, G22, G55, G143, G144, G87, G138, G110, G133, and G120 might be considered as best parents based on the qualitative and quantitative characters especially maximum yield per plant with high PL, PPP, and HSW.” (Lines 584-587)

Comments: 3- The quality of all figures should be improved for publication, the word size and word type should be consistent across all figures. 

Author response: Thank you for your comments. We have changed the word size and type in the manuscript for its consistency.

Round 3

Reviewer 2 Report

I still can see some problems on the figures, eg, strange font type in Fig7.  I recommend to improve the quality of the figures for publication purpose.

Author Response

Reviewer 2 Round 3

Comment: I still can see some problems on the figures, eg, strange font type in Fig7.  I recommend to improve the quality of the figures for publication purpose.

Author response: Thank you for your comment. We have change the font type in Figure 7.
